# Systemic Inflammation in Hip Fracture and Osteoarthritis: Insights into Pathways of Immunoporosis

**DOI:** 10.3390/ijms26189138

**Published:** 2025-09-19

**Authors:** Bernardo Abel Cedeno-Veloz, Alba María Rodriguez-Garcia, Fabricio Zambom-Ferraresi, Soledad Domínguez-Mendoza, Irene Guruceaga-Eguillor, Virginia Ruiz-Izquieta, Juan Jose Lasarte, Nicolás Martinez-Velilla

**Affiliations:** 1Navarre University Hospital (HUN), Navarrabiomed, Institute for Health Research of Navarra (IdiSNA), 31008 Pamplona, Spain; alba.rodriguez.garcia@navarra.es (A.M.R.-G.); fabricio.zambom.ferraresi@navarra.es (F.Z.-F.); soledad.dominguez.mendoza@navarra.es (S.D.-M.); irene.guruceaga.eguillor@navarra.es (I.G.-E.); virginia.ruiz.izquieta@navarra.es (V.R.-I.); nicolas.martinez.velilla@navarra.es (N.M.-V.); 2Heath Sciences Department, Public University of Navarre, 31006 Pamplona, Spain; 3CIBER of Frailty and Healthy Aging (CIBERFES), Instituto de Salud Carlos III, Av Monforte de Lemos, 3-5, Pabellón 11, Planta 0, 28029 Madrid, Spain; 4Immunology and Immunotherapy Program, Centre for Applied Medical Research (CIMA), University of Navarra, IdiSNA, 31008 Pamplona, Spain; jjlasarte@unav.es

**Keywords:** aging, osteoporosis, hip fracture, inflammaging, osteoimmunology, immunoporosis

## Abstract

Inflammaging has been implicated in age-related bone loss and fragility fractures through immune-mediated effects on bone turnover. We aimed to explore the relationship between systemic inflammatory markers and bone health in older adults, focusing on the differences between patients with osteoporotic fractures and non-fractured controls. We retrospectively analyzed 40 older patients (20 with hip fractures and 20 with osteoarthritis without prior fragility fractures). We compared routine inflammatory markers, including red cell distribution width (RDW), C-reactive protein (CRP), neutrophil-to-lymphocyte ratio (NLR), systemic immune-inflammation index (SII), and the composite CRP–albumin–lymphocyte index (CALLY), between groups. Bone mineral density (BMD) at the hip, lumbar spine, and wrist, as well as the FRAX score, were assessed. Correlations between inflammatory markers, BMD, and FRAX scores were evaluated using Spearman’s coefficient. Patients with fractures exhibited significantly elevated CRP (66.2 ± 70.3 vs. 3.8 ± 4.0 mg/L, *p* = 0.0008) and SII (1399.7 ± 1143.4 vs. 751.4 ± 400.8, *p* = 0.025) compared to controls. RDW, NLR, and CALLY scores did not differ significantly between the groups. Higher CRP levels were associated with lower BMD at all sites (hip: r ≈ −0.63, *p* = 0.002; spine: r ≈ −0.60, *p* = 0.005; wrist: r ≈ −0.60, *p* = 0.005). No significant correlations were observed between the SII and BMD or FRAX values. Elevated systemic inflammation, particularly indicated by CRP and SII, was associated with osteoporotic fracture status and low bone density in our cohort. These findings support the concept that inflammatory pathways may contribute to osteoporosis and fracture risk and suggest that inflammatory markers could serve as adjunctive tools in fracture risk assessment. Further studies are required to clarify the causality and evaluate whether targeting chronic inflammation can improve bone health in older adults.

## 1. Introduction

Osteoporosis is a common age-related skeletal disorder characterized by low bone mass and microarchitectural deterioration, leading to increased fragility and fracture risk [1]. The clinical impact of osteoporotic fractures is profound: approximately 50% of women and 20–25% of men over 50 years of age will suffer an osteoporotic fracture, and hip fractures, in particular, carry high mortality and morbidity within one year [2]. Early identification of individuals at high fracture risk is crucial; however, current diagnostic tools have limitations. Dual-energy X-ray absorptiometry (DXA) remains the gold standard for measuring bone mineral density (BMD); however, DXA screening has proven ineffective, and recent studies indicate a weak correlation between BMD and fracture prediction [3]. The Fracture Risk Assessment Tool (FRAX) integrates clinical risk factors (age, sex, prior fractures, comorbidities, etc.) to estimate 10-year fracture probability [4]; however, it does not directly incorporate biomarkers of physiological processes such as chronic inflammation.

A growing body of evidence indicates that inflammaging [5], which refers to the chronic low-grade systemic inflammation that develops with advanced age, can contribute to tissue degeneration (age-related bone) and frailty. Inflammaging is characterized by elevated levels of pro-inflammatory mediators (e.g., interleukins and tumor necrosis factor) in older adults without overt infection, which can alter bone remodeling dynamics. Osteoimmunology [6], the interplay between the immune and skeletal systems, provides a framework for understanding how immune-derived inflammation influences bone metabolism. Inflammatory cytokines stimulate osteoclastogenesis (bone-resorbing cell activity) and suppress osteoblast function, tipping the balance toward bone resorption and reduced bone formation [7]. Over time, this can lead to bone loss, microstructural damage, and an increased susceptibility to fractures. This concept has given rise to the term “immunoporosis”, where osteoporosis is driven or exacerbated by immune system activation [8].

Traditional inflammatory markers and novel immune cell indices are increasingly being studied in the context of bone health [9]. C-reactive protein (CRP), an acute-phase reactant, is a well-established systemic inflammation marker and has been associated with osteoporosis and fractures in epidemiological studies [10]. However, these analyses found only weak correlations or predictive power of CRP for BMD and fracture after accounting for other factors [11]. The neutrophil-to-lymphocyte ratio (NLR), derived from routine blood counts, is an inexpensive marker of systemic inflammation and immune status. Elevated NLR values often indicate an inflammatory milieu and have been linked to poorer outcomes in various chronic diseases, including OP [12]. However, other studies have found no direct correlation between NLR and BMD in cross-sectional analyses, suggesting that their relationship with bone health may be context-dependent [13]. Systemic Immune-Inflammation Index (SII) has been proposed as a more comprehensive measure of systemic inflammatory status. SII combines neutrophil, platelet, and lymphocyte counts into a single index (usually calculated as neutrophils × platelets/lymphocytes) [14]. Elevated SII levels have been associated with lower bone mass and higher odds of osteoporosis in postmenopausal women [14]. However, there is a limited number of available studies, and these findings need to be explored, especially in older adults [9]. Red cell distribution width (RDW), an index of erythrocyte size variability, is another marker linked to inflammation and general health in the older population. A higher RDW has been associated with many age-related conditions, mortality, and fractures (but only in patients without anemia [15]). This suggests that RDW, although not a direct bone marker, may capture aspects of frailty or inflammation relevant to fracture risk [6]. The CRP–albumin–lymphocyte index (CALLY) combines CRP, serum albumin, and lymphocyte count into a single ratio reflecting inflammatory and nutritional status. The CALLY has shown prognostic value in chronic inflammatory diseases [16], but its utility in osteoporosis or fracture risk remains unexplored.

One of the main problems related to all inflammatory markers is their overlap with other conditions related to older adults, specifically osteoarthritis (OA). This overlap between risk factors [17,18] and the inverse relationship between hip fractures and hip OA [19] suggests that developing immunology biomarkers capable of distinguishing between inflammation in bone (OP) and joint (OA) holds promise for improving the diagnosis and prognosis of osteoarticular diseases.

Considering the above, our study aimed to investigate the relationships between systemic inflammatory markers and bone health measures in older adults with hip fractures compared to those with OA. We focused on determining whether inflammatory markers differed by fracture status and whether they correlated with BMD and FRAX. We hypothesized that older adults with a history of osteoporotic fractures would exhibit a higher inflammatory profile and that elevated inflammation would be associated with lower BMD and higher FRAX. Demonstrating such associations could support the use of inflammatory biomarkers as complementary tools in fracture risk assessment and underscore the importance of controlling inflammation in maintaining bone health.

## 2. Results

### 2.1. Patient Characteristics

A total of 40 older adults were included in the analysis and divided into two groups: 20 patients with a recent hip fracture and 20 patients with a diagnosis of osteoarthritis but without prior fractures. As shown in Table 1, patients in the fracture group were significantly older (mean age 87.3 ± 6.7 years vs. 75.2 ± 4.2 years, *p* = 0.026) and had a lower body mass index (BMI) than non-fractured participants (24.9 ± 2.7 kg/m^2^ vs. 29.9 ± 5.0 kg/m^2^, *p* = 0.003).

Functional assessments revealed that the fracture group had significantly worse performance, with lower Barthel Index scores (*p* < 0.001), reduced handgrip strength (*p* < 0.001), higher frailty scores (*p* < 0.001), and more frequent cognitive impairment based on Pfeiffer’s test (*p* < 0.001). Nutritional status, measured using the MNA score, was also significantly poorer in the fracture group (18.8 ± 6.1 vs. 28.0 ± 2.3, *p* < 0.001), with a higher prevalence of depressive symptoms (*p* = 0.026).

Regarding fracture risk, FRAX scores were substantially higher in the fracture group, both for the 10-year major osteoporotic fracture risk (FRAX-T: 13.4 ± 7.0 vs. 6.1 ± 5.3, *p* < 0.001) and hip fracture risk (FRAX-C: 6.3 ± 3.8 vs. 2.6 ± 2.9, *p* < 0.001). In terms of bone mineral density (BMD), patients with hip fractures had significantly lower values at the total hip (*p* = 0.001) and lumbar spine (*p* = 0.007), while the differences at the wrist were not statistically significant.

These group-based distinctions were further visualized via Principal Component Analysis (PCA) (Figure 1), which demonstrated clear clustering of fracture and non-fracture participants with 95% confidence ellipses. The PCA supports that the fracture group differs in multiple clinical and inflammatory domains.

### 2.2. Inflammatory Markers Between Groups

As shown in Table 2, the systemic inflammatory markers were significantly elevated in the fracture group. Specifically, C-reactive protein (CRP) levels were markedly higher among patients with fractures (66.17 ± 70.34 mg/L) than in the nonfracture group (3.80 ± 3.97 mg/L) (*p* < 0.01). Similarly, the systemic immune-inflammation index (SII) was significantly elevated in the fracture group (1399.71 ± 1143.43 vs. 751.41 ± 400.81, *p* = 0.025). No statistically significant differences were observed in red cell distribution width (RDW), neutrophil-to-lymphocyte ratio (NLR), or CALLY between the groups, although trends were noted.

### 2.3. Correlation Between Inflammatory Markers, BMD, and FRAX

In the overall sample, correlation analysis revealed a consistent inverse relationship between CRP and BMD at all measured skeletal sites (Figure 2). CRP levels showed a moderate to strong negative correlation with hip BMD (CAD–BMD), lumbar spine BMD (LUM–BMD), and wrist BMD (WRIST–BMD), with Spearman’s ρ ranging from −0.60 to −0.63 (*p* < 0.01).

However, SII did not show statistically significant correlations with either BMD or FRAX score. Neither CRP nor SII was significantly associated with FRAX-T or FRAX-C in the overall cohort. The full results of the correlation matrix are presented in Appendix A.

## 3. Discussion

In this study, we investigated the interplay between systemic inflammation and bone health in older adults, with particular attention to the differences between those with and without osteoporotic fractures. Our findings provide further evidence that chronic inflammation is associated with osteoporosis and fracture risk in older adults. Specifically, we observed that patients with fragility fractures had significantly higher CRP and SII levels, and that CRP was inversely correlated with BMD at the hip, spine, and wrist. These results support the concept that inflammatory processes, the hallmarks of inflammaging, can adversely affect bone density and contribute to fracture susceptibility [5].

C-reactive protein (CRP) and the Systemic Immune-Inflammation Index (SII) did show differences between our fracture vs. control groups. Our finding is biologically consistent with osteoimmunology [5]. Aging skews bone–immune crosstalk toward chronic low-grade inflammation, where IL-6, TNF-α, IL-1, and IL-17 upregulate RANKL, activate NF-κB/NFATc1, and tip remodeling toward osteoclastogenesis while suppressing osteoblast survival [20]. CRP likely mirrors IL-6-driven acute-phase signaling, whereas SII may reflect age-related myeloid bias and macrophage M1 polarization, both compatible with accelerated resorption and lower BMD [6]. Moreover, IL-6 may play a direct mechanistic role in the pathophysiology of common geriatric diseases related to inflammaging [21]. This suggests they capture inflammatory dynamics more specific to the osteoporotic fracture risk state, as an indirect measure of IL-6 levels, with less interference from osteoarthritis.

One of the most striking results was the markedly elevated C-reactive protein level in the fracture group compared to that in the control group. Elevated CRP levels have been linked to low BMD and greater bone loss in several studies [10] and in osteoarthritis as well [22]. This discrepancy may be attributed to several factors (comorbidities and age were different); however, while CRP can be elevated acutely (e.g., after a fracture or surgery), our retrospective design and data suggest that fracture patients had high CRP chronically or at baseline, indicating an underlying pro-inflammatory state [23]. In our cohort, the stronger correlation may reflect a more pronounced inflammatory influence among the selected patients. This reinforces that CRP is a useful marker of systemic inflammation relevant to bone health, but it may act more as an indicator of an underlying cytokine milieu (for example, IL-6, TNF-α) that directly affects bone remodeling. High CRP levels likely mirror high circulating IL-6 [24], a cytokine known to stimulate osteoclast formation and activity, thereby increasing bone resorption [25].

The systemic immune-inflammation index (SII) was also significantly higher in patients with fractures. In a large NHANES analysis of older adults, it was significantly associated with spine DXA metrics and self-reported osteoporosis status [26]. Neutrophils [27] and platelets [28] participate in acute inflammatory responses and potentially in inflammaging [5]. Neutrophils and platelets can modulate healing processes by releasing growth factors [29]; however, their elevated counts in older adults can also signal inflammation or previous reactive processes related to osteoimmunology [6]. This cascade of immune cells [8], oxidative stress [30], and inflammatory cytokines [31] can dysregulate bone turnover, leading to immunoporosis and fractures [8]. Our finding aligns with a recent meta-analysis where higher SII was associated with OP [9]. However, the marginal OR and study bias limited the results. In our cohort, despite the difference between the fracture and non-fracture groups, we did not detect a significant correlation between the SII and FRAX or BMD. The main reason is that multiple immune cells (including neutrophils and platelets) are also related to OA [32]. These findings highlight the intricate interplay between immune cells, OP, and OA, with overlapping physiopathology [18], suggesting potential complications in differentiating one pathology from another. An elevated SII might indicate an immune-inflammatory imbalance that could contribute to bone loss or reflect poor overall homeostasis in older adults; however, its validation in osteoporosis is still debatable.

Other markers, such as the neutrophil-to-lymphocyte ratio (NLR), red cell distribution width (RDW), and CALLY, did not differ significantly between the fractured and non-fractured groups. A likely explanation is that all these markers can be elevated by both osteoporosis-related frailty and severe osteoarthritis, blunting their discriminatory power. In our study, the “control” patients had advanced osteoarthritis, which itself can spur low-grade inflammation and systemic effects [33]. Therefore, NLR, RDW, and CALLY were high in both groups, failing to distinguish high-fracture-risk osteoporosis from severe osteoarthritis.

Chronic inflammation plays a role in osteoporosis, and higher Neutrophil–Lymphocyte Ratio (NLR) values have been reported in osteoporotic patients [34]. An elevated NLR reflects a shift toward innate immune activation (more neutrophils) with relative lymphopenia, indicating systemic inflammation [35]. However, osteoarthritis also involves inflammatory processes (especially in severe knee or hip OA) [36]. Studies have noted that knee OA patients have higher NLR than healthy controls, but not type or disease severity, suggesting its limited use in indicating OA [37]. The mechanism may involve pro-inflammatory cytokines (IL-6, IL-8, TNF-α) in OA that elevate neutrophils and suppress lymphocyte counts [38]. Thus, in our cohort, both groups likely had moderately elevated NLR due to underlying inflammation. This overlap [18] makes NLR a non-specific indicator—not particularly useful for distinguishing osteoporotic fracture risk from osteoarthritic inflammation. In summary, NLR is a marker of systemic inflammatory status present in both conditions, so any difference was muted.

Prior research links red cell distribution width (RDW) to osteoporotic fracture risk [15]. This suggests that RDW is a frailty/inflammation marker that indirectly tracks bone vulnerability through the effects of general inflammaging to bone marrow [39]. However, RDW is not specific to osteoporosis. Recent population data indicate that RDW is also elevated in osteoarthritis, and it is a significant predictor of OA risk [40]. The likely explanation is that severe OA, especially in older adults, often coexists with systemic factors like chronic inflammation or poor nutrition that can raise RDW [39]. Our control group’s severe OA could thus elevate RDW to a similar range as the fracture group. Both groups in our study likely had similar chronic inflammation/nutrition statuses, yielding overlapping RDW values.

The CALLY is a composite marker that integrates inflammatory burden (C-reactive protein), nutritional status (albumin), and immune status (lymphocyte count). A low CALLY score (meaning high CRP, low albumin, and low lymphocytes) signifies a state of inflammation with malnutrition and immunosuppression. We hypothesized that such a pro-inflammatory [8], poorly nourished state [41] would also adversely affect bone health—inflammation can stimulate osteoclast activity [7], while malnutrition (low protein/albumin) impairs bone formation and increases sarcopenia and falls [42]. However, our data showed no significant CALLY difference between groups. This again likely reflects that CALLY is a biomarker related to OA prediction [43]. Thus, CALLY overlap in our groups underscores that the malnutrition–inflammation syndrome was present in both, limiting its utility for differentiation.

Our results underscore the complex interplay between inflammation and bone. The concept of osteoimmunology [6] predicts that chronic immune activation can lead to bone loss through continuous osteoclast stimulation and inhibition of osteoblasts. The significant CRP-BMD correlations we found align with this, implying that even in subclinical inflammation, bone remodeling is shifted toward resorption [44]. These findings fit into the broader framework of immunoporosis, wherein chronic inflammation is considered a driving factor for osteoporosis progression [8]. It is worth noting that inflammation’s impact on bones might also be partly mediated through muscle and physical function. High systemic inflammation is associated with sarcopenia [45] and frailty [46], which can increase fall risk and thus fracture risk. While the role of these inflammatory indices after hip fracture is associated with worse outcomes, like mortality [47], their role as discriminatory tools between OP, OA, and other rheumatologic diseases remains unclear [48]. Other emerging molecular techniques offer deeper insights into the mechanisms of inflammaging-immunoporosis (such as cytokine profiling, transcriptomics, or advanced omics approaches) [5] and yield a far deeper understanding of disease mechanisms. Although they are necessary, these methods remain resource-intensive, technically demanding, and hard to implement in routine clinical settings. In this context, the use of accessible, blood-based composite indices offers a clinically translatable means to capture the low-grade inflammatory state characteristic of inflammaging.

Clinical relevance and multidimensional strength: Our results advance a pragmatic message for geriatric care: low-cost, routinely available inflammatory indices—especially CRP and SII—distinguished hip-fracture osteoporosis from advanced OA, and CRP tracked lower BMD across sites. Beyond any single marker, our multidimensional assessment (clinical function, nutrition, cognition, site-specific BMD, FRAX, and systemic indices, summarized by PCA) is a key strength, aligning biomarker signals with real-world phenotypes of frailty and fracture [49]. Their integration into fracture risk assessment or frailty screening could help bridge the gap between cutting-edge molecular research and real-world geriatrics practice [50], especially in settings with limited resources. Our findings support this translational approach by demonstrating the discriminatory value of certain markers—particularly SII and CRP—in differentiating patients with high fracture risk from those with advanced osteoarthritis. Although the novelty at the level of “inflammation and bone” is incremental, composite indices (SII, CALLY) have not been systematically studied in hip-fracture populations. Our pilot data thus guide the next steps: (i) multivariable models adding CRP/SII to FRAX, (ii) validation of clinically interpretable cut-offs (ROC), and (iii) external replication across settings.

Limitations: Several limitations of our study should be acknowledged. First, the sample size was modest (n = 40), which limits statistical power and the generalizability of the findings. However, the exploratory nature of this study still provides valuable, hypothesis-generating insights. Second, the retrospective design prevented us from standardizing the timing of blood sample collection in relation to fracture events or clinical stability, potentially affecting the levels of inflammatory markers. Third, our control group was composed of patients with advanced osteoarthritis rather than healthy individuals, which may have introduced a confounding inflammatory background and reduced the contrast between groups, but it is crucial to establish a different approach to these two major skeletal diseases in older adults. Although we set an a priori inclusion threshold of ≥70 years, the fracture cohort was older than the osteoarthritis controls. This difference mirrors the real-world epidemiology in our population—hip fractures typically occur in the late 80s, whereas advanced hip OA candidates for arthroplasty are commonly in their mid-70s [51]. As an exploratory, hypothesis-generating pilot, our comparisons should be interpreted with caution. Despite these limitations, the study contributes to the growing body of evidence on the relevance of immunological markers in bone health and aging and highlights the potential utility of pragmatic, accessible inflammatory indices in clinical practice.

Future Directions: Further research involving larger, prospective cohorts is warranted to confirm and expand upon these findings. Longitudinal studies are particularly needed to evaluate whether baseline levels of CRP, SII, NLR, and other systemic inflammatory markers can predict subsequent bone loss, decline in functional status, or incident fractures. Although our study was not designed to address therapeutic effects, the observed associations reinforce the rationale for considering inflammation as a modifiable contributor to osteoporosis [52]. Future studies will recruit age-matched cohorts and apply multivariable adjustment to determine the independent contribution of systemic inflammatory indices beyond age. They should also include OA participants to avoid the overlap in the inflammatory signature between OP and OA patients [18].

In addition, future investigations should assess whether integrating systemic inflammatory markers into existing fracture risk models (e.g., by incorporating CRP or SII into FRAX) improves predictive accuracy and risk stratification. Such enhanced models could support more personalized prevention strategies. Another promising avenue involves the integration of clinical, biochemical, and imaging data with advanced machine learning approaches—particularly deep learning models (DLMs) [53]—to develop scalable, accurate tools for predicting fracture risk across diverse and aging populations. This multidisciplinary direction, bridging clinical assessment and artificial intelligence, may facilitate the early identification of high-risk individuals and support the implementation of preventive strategies in real-world settings.

Future work should prioritize prospective validation of composite blood–cell indices in diverse hip-fracture cohorts, define cut-offs linked to outcomes (fractures, function, mortality), and test integration with FRAX to quantify incremental value. Where feasible, parsimonious feature-selection can fuse inflammatory indices with site-specific BMD and geriatric covariates, preserving interpretability and facilitating clinical adoption.

## 4. Materials and Methods

Study Design and Patients: We conducted a retrospective observational study of older adult patients who were already part of a previous study [25].

Participants: Eligibility, Identification: The criteria for participant inclusion were age  ≥  70 years, who were divided into two real-world geriatric phenotypes. We identified two groups: (i) hip-fracture group (n = 20): older adults admitted with low-trauma osteoporotic hip fractures (fall from standing height or less), including femoral neck or intertrochanteric fractures, confirmed by orthopedic and radiographic assessment. Fragility fractures were defined as low-trauma fractures occurring at standing height or less, consistent with osteoporosis. (ii) Osteoarthritis (OA) group (n = 20): consisted of older adults with advanced hip osteoarthritis (OA), identified by their orthopedic teams as candidates for elective hip arthroplasty, with clinical and radiographic evidence of OA documented in the medical record. OA was defined by the American College of Rheumatology [54] criteria. Exclusion criteria included secondary osteoporosis (e.g., long-term glucocorticoids, rheumatoid arthritis, autoimmune diseases), active malignancy or end-stage disease, and lack of sufficient clinical/laboratory/DEXA data. Selection and classification were performed by two investigators independently; discrepancies were adjudicated by a senior geriatrician.

Clinical, Functional, and Nutritional Assessment: We collected age, sex, BMI, comorbidities (CIRS–G), polypharmacy, and a physician-diagnosed osteoporosis label, and we used the following functional measures: Barthel Index (ADL), Functional Ambulation Category (FAC), handgrip strength, frailty score, Mini-Nutritional Assessment (MNA), Pfeiffer’s SPMSQ, and Geriatric Depression Scale-15.

Bone Densitometry and FRAX: For each patient, we collected demographic data (age and sex) and clinical information relevant to bone health (including any diagnosed osteoporosis, comorbid conditions, and current medications). We retrieved bone mineral density (BMD) measurements at three skeletal sites: the proximal femur (hip BMD), lumbar spine (L1–L4 BMD), and distal radius (wrist BMD), measured by DXA (Lunar iDXA, GE Healthcare); BMD values were recorded in g/cm^2^ [55]. We also obtained each patient’s fracture risk assessment (FRAX), specifically the 10-year probability of a major osteoporotic fracture (“FRAX-Total”) and of a hip fracture (“FRAX-Hip”) [56].

Timing of Assessments: Clinical evaluation of the patients was performed on the day of the surgery. Inflammatory markers were obtained as part of routine care 6–12 months before surgery (morning draws, fasting for the metabolic panel). DXA was performed approximately 1 month after surgery using the same device and protocol for all patients.

Inflammatory Marker Assessment: these included the following:

C-reactive protein (CRP): measured in serum (mg/L) using a standard immunoassay as a marker of systemic inflammation.

Red cell distribution width (RDW): part of the complete blood count, expressed as a percentage (%), indicating the variability in red blood cell size.

Neutrophil-to-lymphocyte ratio (NLR): calculated from the differential white cell count as the absolute neutrophil count divided by the lymphocyte count.

Systemic immune-inflammation index (SII): calculated as (absolute neutrophils × absolute platelets)/absolute lymphocytes.

CRP–albumin–lymphocyte index (CALLY): computed as (serum albumin × lymphocyte count)/(CRP × 10).

All blood samples were collected in the morning (after an overnight fast for the metabolic panel) as part of routine evaluation. Standard automated analyzers were used for blood counts and biochemistry in the Laboratorio Unificado de Navarra (LUNA), ensuring consistency across patients.

Statistical Analysis: We first compared the fractured and nonfractured groups in terms of inflammatory marker levels and other key variables. Continuous variables are expressed as mean ± standard deviation (SD). Given the small sample size per group (n = 20), we primarily used non-parametric tests: Mann–Whitney U for between-group comparisons and Spearman’s ρ for correlations. Normality was screened with Shapiro–Wilk; results based on parametric tests (Student’s *t*, Pearson’s *r*) were relegated to sensitivity checks, yielding consistent conclusions. We report effect sizes (rank-biserial correlation for Mann–Whitney; ρ for Spearman) with 95% CIs.

Principal component analysis (Figure 1) assessed the distribution groups, using singular value decomposition with imputation (pre-normalized data, no transformation), and visualized using ClustVis [57].

We evaluated the correlations between statistically significant inflammatory markers, BMD, and FRAX using Spearman’s correlation coefficient, and significant differences between the groups were used to assess performance. A correlation matrix was created using the SRPLOT online platform (https://www.bioinformatics.com.cn/en, accessed on 22 February 2025). Given the pilot and retrospective nature, no a priori sample size was calculated; findings are considered exploratory and hypothesis-generating.

## 5. Conclusions

In conclusion, our study reinforces that systemic inflammatory markers, notably CRP and SII, are elevated in older adults with osteoporotic fractures, and that higher inflammation is associated with lower bone density. These findings support the biological link between chronic inflammation and skeletal degeneration during aging. Recognizing the inflammaging–immunoporosis connection has potential clinical implications: routine inflammatory markers may help identify high-risk individuals or serve as therapeutic targets to preserve bone health. Our work adds a piece to the puzzle of osteoimmunology, emphasizing that bone health in older adults is not only a matter of hormones and calcium but also of the immune system and inflammation status. Managing systemic inflammation may thus emerge as a novel adjunct strategy in combating osteoporosis and reducing fracture risk in our aging population.

## Figures and Tables

**Figure 1 ijms-26-09138-f001:**
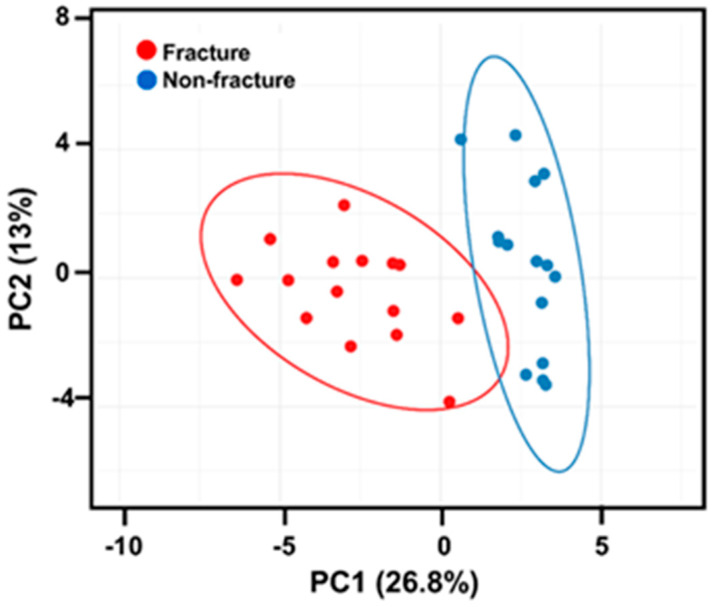
Principal component analysis (PCA) of the study groups. The ellipses show a probability of 95% that a new data point from the same group is located inside the ellipse (red points—fracture participants; blue points—non-fracture (OA) participants).

**Figure 2 ijms-26-09138-f002:**
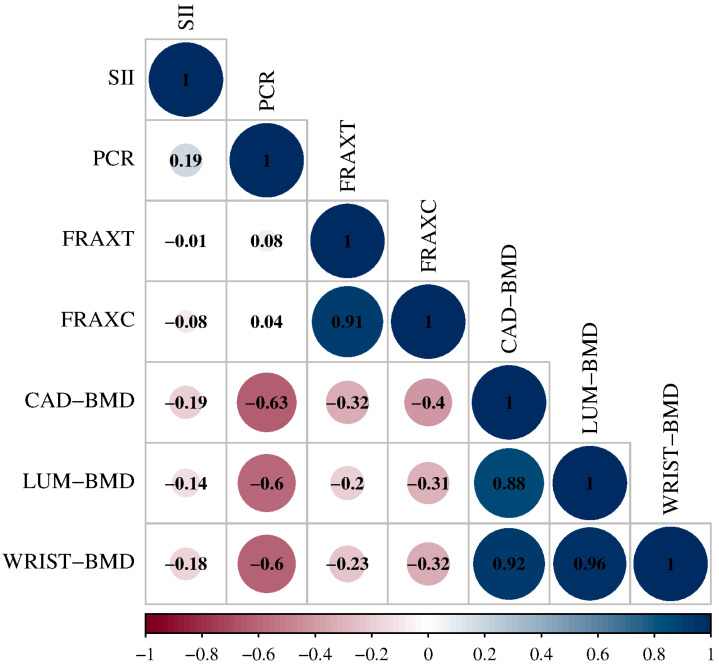
Correlation matrix among inflammatory markers, FRAX scores, and BMD. Spearman’s ρ with circle size/color reflects correlation magnitude/direction (blue = positive, red = negative). Abbreviations: SII, systemic immune-inflammation index; CRP, C-reactive protein; FRAX-T, 10-year probability of a major osteoporotic fracture; FRAX-C, 10-year probability of hip fracture; CAD–BMD, total hip BMD; LUM–BMD, lumbar spine BMD; WRIST–BMD, distal radius BMD.

**Table 1 ijms-26-09138-t001:** Demographic, clinical, and functional characteristics of the patients included in the analysis.

	Fracture Group (n = 20)	Non-Fracture (OA) Group (n = 20)	*p* Value *
** *Demographic* **			
Age, years	87.25 (6.73)	75.20 (4.15)	**0.026**
Sex (men/female), n (%)	4 (20)/16 (80)	7 (35)/13 (65)	0.480
BMI (kg/m^2^)	24.91 (2.74)	29.87 (5.02)	**0.003**
** *Clinical status* **			
CIRS-G score	12.7 (4.81)	10.2 (3.17)	0.060
Polypharmacy score	7.25 (3.09)	5.3 (3)	0.534
Osteoporosis (n, %)	4 (20%)	6 (30%)	0.716
** *Functional status* **			
Barthel Index (ADL), score	67.5 (30.41)	95.75 (7.48)	**<0.001**
Functional Ambulation Category (n, %)			
FAC 0 to 1	3 (15%)	0 (0)	**0.032**
FAC 4 to 5	17 (85%)	20 (100%)	
Frailty score	3.05 (1.47)	1.3 (1.42)	**<0.001**
Hand grip strength (kg)	11.3 (6.24)	23.95 (8.6)	**<0.001**
MNA score	18.83 (6.08)	28.03 (2.33)	**<0.001**
Pfeiffer’s SPMSQ	5.05 (4.05)	0.5 (0.224)	**<0.001**
Depression score (n, %)	6 (42.9%)	2 (10%)	**0.026**
FRAX major score	13.4 (6.99)	6.12 (5.29)	**<0.001**
FRAX hip score	6.29 (3.79)	2.58 (2.94)	**<0.001**
* **Bone mineral density and body composition** *			
BMD—total hip	0.735 (0.079)	0.976 (0.177)	**0.001**
BMD—lumbar spine	0.981 (0.18)	1.239 (0.247)	**0.007**
BMD—wrist	0.679 (0.127)	0.812 (0.37)	0.281

* *p*-value for different groups in percentage (Fisher’s exact test) or means (U de Mann–Whitney). The bold values are statistically significant. Abbreviations: BMI (body mass index); CIRS-G, Cumulative Illness Rating Scale for Geriatrics, which evaluates individual body systems, ranging from 0 (best) to 56 (worst); Barthel Index, which ranges from 0 (severe functional dependence) to 100 (functional independence); Frail Scale, which ranges from 0 to 5 and indicates frailty with ≥3; Mini-Nutritional Assessment (MNA); Pfeiffer’s Short Portable Mental State Questionnaire (SPMSQ), which ranges errors from 0 (best) to 10 (worst); The Geriatric Depression Scale (GDS-15), which ranges from 0 to 15 and indicates symptomatic depression with ≥5; FRAX, 10-year fracture probability of major osteoporotic fracture (%). Mean and SD; FRAX 10-year fracture probability of hip fracture (%); BMD (bone mineral density, g/cm^2^).

**Table 2 ijms-26-09138-t002:** Comparison of systemic inflammatory markers between the fracture and non-fracture (OA) groups.

	Fracture (Mean, SD)	Non-Fracture ((OA) Mean, SD)	*p*-Value
RDW	14.66 (1.47)	26.67 (38.17)	0.18
CRP	66.17 (70.34)	3.80 (3.97)	**<0.01**
NLR	6.76 (4.84)	4.80 (2.34)	0.11
SII	1399.71 (1143.43)	751.41 (400.81)	**0.025**
CALLY	0.97 (1.94)	4.52 (9.92)	0.13

Comparison of systemic inflammatory markers between fracture and non-fracture groups. Data are mean (SD). *p*-values from Mann–Whitney. The bold values are statistically significant. Abbreviations: RDW, red cell distribution width; CRP, C-reactive protein; NLR, neutrophil-to-lymphocyte ratio; SII, systemic immune-inflammation index; CALLY, CRP–albumin–lymphocyte index.

## Data Availability

All data relevant to the study are included in the article, have already been published, or uploaded as Appendix A.

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
