# Peer review of "Systemic Inflammation in Hip Fracture and Osteoarthritis: Insights into Pathways of Immunoporosis"

_ijms, 2025, doi:10.3390/ijms26189138_

Round 1

Reviewer 1 Report

Comments and Suggestions for Authors

An interesting experimental study,  in which the authors address an important and clinically relevant question: the relationship between systemic inflammation and osteoporotic fractures.This study provides valuable pilot evidence that systemic inflammation, especially CRP, is linked to osteoporosis and fracture status. While limited by size and cross-sectional design, it highlights inflammation as a promising adjunctive factor in fracture risk assessment and opens avenues for preventive and therapeutic strategies.

There are following suggestions for this work:

  1. The methodology section is very brief , more detailed information must be added.
  2. Extensive research has already been done on inflammation in bone health ,so the novelty of this work is average. However, since composite indices of systemic inflammation (SII and CALLY), have not been systematically evaluated in osteoporotic fracture populations. Reframing the discussion section to highlight the clinical relevance of the findings as well as the importance of the research outcome to serve as a guide for future biomarker research, would make the work more interesting to the readers. The multi-dimensional assessment done should be exhibited as the strength of this study.

minor Comments:

The legend for Table 2 and Figure 2 are too long , observations made here need not be mentioned in the legend. 

Author Response

Dear Reviewer 1,

We sincerely thank the reviewer for the thoughtful and constructive feedback. We have revised the manuscript accordingly. Below we address each point in detail.

General comment

An interesting experimental study, in which the authors address an important and clinically relevant question: the relationship between systemic inflammation and osteoporotic fractures. This study provides valuable pilot evidence that systemic inflammation, especially CRP, is linked to osteoporosis and fracture status. While limited by size and cross-sectional design, it highlights inflammation as a promising adjunctive factor in fracture risk assessment and opens avenues for preventive and therapeutic strategies.

Response: We appreciate the positive assessment. Our revisions aim to strengthen clinical relevance, transparency of methods, and the translational message for biomarker research.

1)“The methodology section is very brief; more detailed information must be added.”
Response: We expanded Materials and Methods substantially to enhance transparency and reproducibility

2) “Extensive research has already been done on inflammation in bone health ,so the novelty of this work is average. However, since composite indices of systemic inflammation (SII and CALLY), have not been systematically evaluated in osteoporotic fracture populations. Reframing the discussion section to highlight the clinical relevance of the findings as well as the importance of the research outcome to serve as a guide for future biomarker research, would make the work more interesting to the readers. The multi-dimensional assessment done should be exhibited as the strength of this study.”

Response: We revised the Discussion to:

  • Emphasize clinical relevance: low-cost, routinely available indices (especially CRP and SII) as adjuncts to BMD/FRAX in older adults.
  • Explicitly present our multi-dimensional assessment (clinical function, nutrition, cognition, site-specific BMD, FRAX, inflammatory indices, PCA) as a key strength that enhances translational value.
  • Clarify the applied novelty: composite indices (SII, CALLY) have not been systematically evaluated in hip-fracture populations; our pilot provides a foundation to guide future biomarker research (prospective validation, multivariable models augmenting FRAX, and clinically interpretable cut-offs via ROC).
    New paragraphs have been inserted in the Discussion and Future Directions to reflect these points.

Minor comment: “The legend for Table 2 and Figure 2 are too long; observations made here need not be mentioned in the legend.”

Response: We shortened the legends for Table 2 and Figure 2 to include only essential information and abbreviations, moving interpretative text into the Results. The revised legends now comply with IJMS style.

We hope these changes address the reviewer’s concerns and improve the clarity and impact of the manuscript. We are grateful for the helpful suggestions.

Sincerely,

The Authors

Reviewer 2 Report

Comments and Suggestions for Authors

Systemic inflammation acts as both a facilitator and a consequence of degenerative processes in the hip region. In hip fractures, it primarily contributes to reduced bone strength and delayed healing. In hip osteoarthritis, it exacerbates cartilage degradation and symptom severity. Understanding these inflammatory pathways offers potential targets for preventive and therapeutic strategies aimed at reducing fracture risk and slowing OA progression. Therefore, the study of Dr. Cedeno-Veloz et al is very important.

Comments

  1. All the typos should be corrected. All the repeated statements should be deleted.
  2. Line 22: The authors should indicate that non-fractured controls were osteoarthritis patients. This should be corrected.
  3. Line 29, 388-394: If the study groups is less than 30 subjects, non-parametric statistical methods should be applied. This should be corrected.
  4. Lines 18, 60, 233: The definition of inflammaging involved indication the low-grade inflammation that develops with advanced age. Multiple definitions should be avoided. This should be corrected.
  5. Table 1: Column 2 should be deleted and all the information should be described in the Section 2.1. This should be corrected.
  6. Table 1: It is scientifically incorrect to compare groups of patients which are significantly different according to their age. This should be corrected.
  7. Table 2: Column 2 should be deleted. This should be corrected.
  8. Figure 2: The method used for correlation analyses should be indicated in the Figure caption.
  9. In the Discussion the authors should describe in detail the biomarkers which demonstrated significant importance. Other biomarkers, such as NLR, RDW, CALLY should be only mentioned while all the detailed information about these markers should be moved to the Introduction. This should be corrected.
  10. Line 302: Molecular techniques should be used in modern rheumatology although it requires higher level of education and skills. Because it gives much more profound understanding of the diseases processes.
  11. Line 321: The involvement of osteoarthritic patients in the control group should be indicated in the Figure and Table captions.
  12. Line 354: The composition of the examined groups is not clear in terms of osteoarthritis disease. The authors should clearly describe the patients of both groups related to their illnesses. This should be corrected.

Author Response

Dear Reviewer 2,

We are grateful for your insightful and supportive comments. We have implemented the changes below, which we believe improve clarity, methodological rigor, and clinical relevance.

General comment

Systemic inflammation acts as both a facilitator and a consequence of degenerative processes in the hip region. In hip fractures, it primarily contributes to reduced bone strength and delayed healing. In hip osteoarthritis, it exacerbates cartilage degradation and symptom severity. Understanding these inflammatory pathways offers potential targets for preventive and therapeutic strategies aimed at reducing fracture risk and slowing OA progression. Therefore, the study of Dr. Cedeno-Veloz et al is very important.

Response: We appreciate the positive assessment. Our revisions aim to strengthen clinical relevance, transparency of methods, and the translational message for biomarker research.

  • All the typos should be corrected. All the repeated statements should be deleted.

Response: We carefully proofread the manuscript, corrected typographical errors, and removed repeated statements to streamline the narrative.

  • Line 22: The authors should indicate that non-fractured controls were osteoarthritis patients. This should be corrected.

Response: We now explicitly state in the Introduction and Methods that the non-fracture control group consists of patients with advanced hip osteoarthritis (OA). We also added this information to table and figure captions.

  • Line 29, 388-394: If the study groups is less than 30 subjects, non-parametric statistical methods should be applied. This should be corrected.

Response: We revised the Statistical Analysis section to only explain the non-parametric methods used in the analysis. The text, tables, and the Figure 2 caption have been updated accordingly.

  • Lines 18, 60, 233: The definition of inflammaging involved indication the low-grade inflammation that develops with advanced age. Multiple definitions should be avoided. This should be corrected.

Response: We retained one concise definition of inflammaging and removed redundant definitions.

  • Table 1: Column 2 should be deleted and all the information should be described in the Section 2.1. This should be corrected.

Response: We deleted the overall (‘full sample’) column from Table 1 as requested.

  • Table 1: It is scientifically incorrect to compare groups of patients which are significantly different according to their age. This should be corrected.

Response: We appreciate this important point. The age difference between groups reflects the real-world phenotypes we aimed to study in our population. In our setting, hip fracture typically occurs in very old adults (often late 80s), whereas advanced hip osteoarthritis candidates for arthroplasty are usually younger within the older-adult spectrum (commonly mid-70s). To ensure we evaluated older adults, we set an a priori inclusion threshold of ≥70 years for both groups.

Our intention was not to claim age-matched causal effects, but rather to provide exploratory, hypothesis-generating insights into how routinely available inflammatory indices behave in these two clinically distinct geriatric phenotypes. We have now made this exploratory nature explicit in the Methods, and we have added the age difference as a limitation in the manuscript, stating that future studies will implement age-matched designs and multivariable adjustment to isolate the independent contribution of inflammatory indices.

We hope this clarifies our rationale and the scope of inference for this pilot study.

  • Table 2: Column 2 should be deleted. This should be corrected.

Response: We deleted the overall column from Table 2 and retained fracture vs. OA with p-values

  • Figure 2: The method used for correlation analyses should be indicated in the Figure caption.

Response: The Figure 2 caption now specifies Spearman’s ρ (non-parametric) as the correlation method.

  • In the Discussion the authors should describe in detail the biomarkers which demonstrated significant importance. Other biomarkers, such as NLR, RDW, CALLY should be only mentioned while all the detailed information about these markers should be moved to the Introduction. This should be corrected.

Response: We thank the reviewer for this thoughtful suggestion. Our study was deliberately designed with advanced hip osteoarthritis as the non-fracture comparator, as explained in the Introduction, to mirror the real-world clinical scenario in which fracture osteoporosis is frequently contrasted with OA in older adults. Because OP and OA share a chronic, low-grade inflammatory milieu, composite “chronic” indices such as NLR, RDW, and CALLY may be elevated in both conditions, thereby reducing between-group contrasts. For this reason, we think that we kept the interpretive note about these non-significant indices in the Discussion to make the overlap explicit and to clarify why they do not discriminate well when OA is the comparator, compared to previous studies.

Moreover, we expanded “Future Directions” to state that future studies should retain OA as a comparator and deploy overlap-sensitive designs so that biomarker performance reflects the true clinical differential of fracture osteoporosis versus advanced OA.

We believe these changes clarify and focus with the translational message that negative or non-discriminative findings are informative and the important of the overlap between these two pathologies.

  • Line 302: Molecular techniques should be used in modern rheumatology although it requires higher level of education and skills. Because it gives much more profound understanding of the diseases processes.

Response: We fully agree that molecular techniques are invaluable in modern rheumatology and provide deeper mechanistic insight. In this pilot, however, our aim was translational and pragmatic—to determine whether low-cost, routinely available indices (CRP and SII) can capture clinically meaningful inflammation in fracture-prone older adults when contrasted with advanced osteoarthritis, thereby facilitating immediate bedside adoption. We have revised the manuscript to make this focus explicit and to emphasize a stepped model: accessible, routine indices as a first-tier for screening and risk stratification, with molecular assays serving as a second-tier for mechanistic phenotyping or equivocal scenarios, and for prospective evaluation of incremental value. The revised paragraph (Discussion) clarifies this complementary role and our rationale for prioritizing clinical feasibility in geriatric practice.

  • Line 321: The involvement of osteoarthritic patients in the control group should be indicated in the Figure and Table captions.

Response: We updated all relevant captions to explicitly indicate that the control group consists of OA patients.

  • Line 354: The composition of the examined groups is not clear in terms of osteoarthritis disease. The authors should clearly describe the patients of both groups related to their illnesses. This should be corrected.

Response: We clarified group composition in the Material section

We appreciate your constructive feedback, which has helped us substantially improve the manuscript.

Sincerely,

The Authors

Round 2

Reviewer 2 Report

Comments and Suggestions for Authors

I have no more comments. Accept as is.